# How does a social norms-based intervention affect behaviour change? Interim findings from a cluster randomised controlled trial in Odisha, India

Erica Sedlander ![ORCID],[1] Ichhya Pant,[2] Jeffrey Bingenheimer,[2] Hagere Yilma,[3] Lipika Patro,[4] Satyanarayan Mohanty,[5] Rohini Ganjoo,[6] Rajiv Rimal[7]

For numbered affiliations see end of article.

**Correspondence to**
Dr Erica Sedlander;
erica.sedlander@ucsf.edu

## ABSTRACT

**Background** Behaviour change interventions targeting social norms are burgeoning, but researchers have little guidance on what they look like, and which components affect behaviour change. The Reduction in Anaemia through Normative Innovations (RANI) project designed an intervention to increase iron folic acid (IFA) consumption in Odisha, India.

**Objective** This paper examines the effect of the intervention at midline to understand which components of the RANI intervention affect uptake.

**Methods** Using a cluster randomised controlled design, we collected baseline data and midline data 6 months later from women of reproductive age in the control and treatment arms (n=3800) in Angul, Odisha, India. Using nested models, we analysed data from three different intervention components, monthly community-based testing for anaemia, participatory group education sessions, and videos, to determine the extent to which exposure to each of these components accounted for the overall intervention effect on haemoglobin and self-reported IFA use.

**Results** Overall, residing in a treatment as opposed to control village had little effect on midline haemoglobin, but increased the odds of taking supplements by 17 times. Exposure to each of the intervention components had a dose–response relationship with self-reported IFA use. These components, separately and together, accounted for most of the overall effect of treatment assignment on IFA use.

**Conclusions** All intervention components increased iron supplement use to differing degrees of magnitude. It appears that a social norms-based approach can result in improving IFA uptake, though improvements in haemoglobin counts were not yet discernible.

## INTRODUCTION

Social norms, defined as informal rules of behaviour considered acceptable in a group or society,[1] are increasingly recognised as drivers of or barriers to behaviour change. In a review of social norms-based interventions over the last three decades, more than half were

published in the last decade.[2] Recent social norms interventions in low-income to middle-income countries have focused on changing harmful behaviours, such as intimate partner violence,[3 4] female genital cutting[5] and child marriage.[6] Past research shows that people are more likely to engage in a behaviour when they believe many others also do so, and when there is a social expectation that they themselves should comply. In the anaemia prevention context, recent literature indicates a seemingly simple behaviour, for example, taking a weekly iron supplement, is embedded in social and cultural dynamics that dictate which health behaviours are appropriate.[7 8]

The theory of normative social behaviour (TNSB) highlights the critical role that social norms can play in influencing health behaviours and the circumstances under which they may do so.[9 10] According to the TNSB, the effects on behaviour of descriptive norms (ie, one's perceptions about others' behaviours) and injunctive norms (ie, one's perceptions of social expectations regarding the behaviour) may depend on other factors at the individual, behavioural, and environmental level.[11]

Factors that could either augment or attenuate the effect of social norms messaging on iron folic acid (IFA) use include perceptions that one is at risk of getting anaemia, outcome expectations that taking IFA will result in health benefits like preventing anaemia or reducing fatigue, and access to and availability of IFA. Therefore, interventions must aim to improve both social norms around a behaviour along with potential moderators that may strengthen the relationship between social norms and behaviours. In a 2018 meta-analysis of social norms approaches to behaviour change, Dempsey *et al* found that the most effective social norms manipulations take place in one's own environment (eg, a field trial), those that deliver messages in multiple formats, and those that target collectivist groups.[12]

Many social norms-based interventions are often complex and resource intensive. To implement them well, programme planners need to understand the social context, the social norms around the behaviour of interest, and barriers and facilitators at multiple levels of the socio-ecological model.[13 14] Guided by the TNSB, the Reduction in Anaemia through Normative Innovations (RANI) Project conducted formative research to delineate significant facilitators to and barriers of IFA consumption in Odisha, India.[8] Findings from the formative work led to the design of the intervention, which included three components: (1) hands-on participatory learning modules conducted in small groups, (2) dissemination of short videos focusing on iron consumption norms and (3) monthly haemoglobin testing for anaemia, followed by public display of (anonymised) community results.

In this paper, we seek to determine which intervention components impact IFA consumption 6 months later at midline. This knowledge allowed us to adapt the intervention implementation according to empirical findings. Since social and political realities on the ground cannot be fully predicted at the outset of any given field trial,[15] we deemed this approach more preferable than simply implementing a static intervention based on a priori data. Thus, the primary goal of this paper is to examine the extent to which each intervention component contributed to the overall effects of the intervention. In addition, because the influence of the intervention may be different across subgroups, we also examine how susceptibility to intervention impact varied by age, caste, education, and communication activity. We hope that delineating the effects of each intervention component will provide guidance for future social norms-based intervention designs. As Davis *et al* ((69), p. 2218) argue, 'while the social norms approach is based in a rich theory, the theory does little to illuminate implementation details of interventions.[16,]

## METHODS
### Patient and public involvement statement
Key stakeholders from the community where the intervention was implemented participated in a 3-day convening in Bhubaneswar, Odisha to co-design the intervention. Community health workers helped to implement the intervention and disseminate findings back to the community. Patients were not involved in the study—participants were living in the villages where the intervention took place.

### Study setting
Nestled in the eastern coast of India, the state of Odisha is predominantly rural. Most residents (95%) are Hindu, with 23% belonging to specific tribes and practising tribal culture (NFHS 2015–2016). At approximately 2.1 children per woman, the total fertility rate sits fairly low. Our focal district, Angul, 1 of 30 in the state of Odisha, has almost 2000 villages with a total population of just over 1.2 million (Government of Odisha, 2019). Men's literacy rate (87%) is higher than that of women's (70%). Almost a quarter of girls (22%) marry before age 18 and around half of married women of reproductive age use modern methods of family planning.[17]

### Intervention development
To develop the RANI intervention, we conducted formative research that examined social norms around IFA use. Between March and May 2018, we collected data from four villages in the two adjacent blocks (administrative units below the district) where the intervention took place (Kishorenagar and Athamalik). We conducted 16 focus groups and 21 individual interviews (n=148), stratified by age and gender, with women of reproductive age, husbands, mothers-in-law, and key informants. To explore women's social norms within the focus groups, we used vignettes, short stories about theoretical characters that also live in a rural village in Angul, India.[18] Vignettes can also help uncover if social sanctions exist and unpack existing social norms. Four researchers, two from India and two from the USA, analysed transcripts using NVivo V.12 to identify barriers and facilitators to IFA use.

We found that social norms and available services varied substantially for pregnant women, non-pregnant women, and adolescents. Specifically, we found that most participants believed only pregnant women and adolescents in school consume IFA (descriptive norms). Participants also stated that only pregnant women and those diagnosed with anaemia should be taking IFA regularly (injunctive norms) and we found that front-line health workers only distributed IFA to pregnant women. Adolescents enrolled in school can also obtain them weekly.[8]

Non-pregnant women (our sample for this paper) were not receiving IFA from front-line health workers. Indeed, barriers faced by non-pregnant women were significant: they needed to visit a health centre, get tested for anaemia, and then obtain the IFA if they were diagnosed as anaemic. We also found that risk perception was low, with most participants believing that only 'a handful of women' in their community had anaemia when, in reality, more than half of women were anaemic. When anaemia was referenced, we found that participants were primarily referring to severe anaemia, not its mild or moderate forms.[8]

Our findings also revealed that inequitable gender norms were an upstream barrier to women's accessing and adhering to IFA supplements. Specifically, women prioritised their family's health and well-being over their own, normalised fatigue as part of a woman's plight, and given that they often do all of the household work while also working outside of the home, they lacked time (and often autonomy) to visit a health centre on their own.[19 20]

We used findings from the qualitative research, past literature on anaemia reduction efforts, and the TNSB to design the RANI intervention. A 3-day convening in Bhubaneswar, Odisha, was held where we invited front-line health workers from the community, anaemia researchers, and programme planners to c-odesign an effective social norms-based intervention. Finally, we used quantitative data collected at baseline to refine the intervention design. Specifically, we validated qualitative findings that very few non-pregnant women were taking IFA with a specific percentage (less than 3% of women) despite guidelines that all women should take them regularly to prevent anaemia.[21] Therefore, we decided to focus more on injunctive norms messaging (that all women of reproductive age should take IFA) rather than descriptive norms messaging (that women are taking IFA) in the beginning of the intervention.

The RANI Project intervention comprises three main components, each tapping into social norms differently: participatory learning modules; RANI Comms (videos); and community haemoglobin testing. A full description of all RANI intervention activities including specifics about each component can be found in box 1. All RANI project data is stored in an online data repository.[22]

## Study design

Data for this study report interim midterm findings from the main trial of the RANI project.[23] The RANI project uses a cluster randomised controlled trial to evaluate the efficacy of a norms-based intervention to increase IFA use and reduce anaemia in Odisha, India. The RANI project selected two blocks within the Angul district in the state of Odisha. We grouped contiguous villages in these blocks into clusters, which we then randomly assigned to either the treatment or control arm. Villages in the treatment arm were exposed to the RANI intervention, while villages in the control arm continued with 'care-as-usual'. We created clusters to minimise contamination; clusters were separated by either a natural buffer (ie, mountain or river) or a village that was neither in the treatment nor in the control arm. This process resulted in a total of 89 clusters from 239 villages. We then segmented clusters by the proportion of caste/ethnic groups (in India they are called scheduled castes and scheduled tribes) and then selected three per stratum, for a total of 15 clusters per arm to be included in data analysis (which comprised 81 villages).

In this paper, we report results from the baseline and midterm assessment, which is a longitudinal study from both the treatment and control clusters. The response rate for the midline questionnaire was 96.2%. Interviewers visited homes up to three times and the primary

reason for not taking the midline survey was not being home when the interviewers visited their house.

## Participants

In each designated village, we first enumerated all households and then randomly selected households for data

---

### Box 1  Description of Reduction in Anaemia through Normative Innovations (RANI) intervention activities

**Group participatory learning sessions**

We developed ten 1 hour-long group participatory learning modules on various topics related to anaemia prevention, including iron folic acid (IFA) supplementation, diet diversity, and social norms/gender norms that may impact a woman's ability to take iron supplements. These monthly group participatory learning sessions are delivered through in-person activities and games. Women and their social networks (eg, husbands and mothers-in-law) were all invited to participate so that women and those important to her are being exposed to the same messaging. We covered four participatory learning modules before midline data collection.

**RANI Comm videos**

We created four RANI Comm videos in the local language, Odiya, with local residents as actors. These videos highlighted the stories of women overcoming barriers related to IFA consumption and other social norms prevalent in the area. The videos were shown on smartphones and tablets to both individuals and small groups. The videos targeted various audiences (pregnant women, non-pregnant women, husbands, and mothers-in-law) and addressed barriers and facilitators to IFA use that we identified in the formative research followed by group discussion sessions. Descriptive and injunctive norms messages around IFA consumption were included in each storyline. An example of injunctive norms messaging in the storyline is a mother-in-law expressing that her daughter-in-law should be taking care of herself too, not just looking after the family, and should be taking weekly IFA to avoid anaemia even if she is not pregnant. A new video was rolled out every 2–3 months. The viewing time for one video was approximately 15 min. Even though videos were 3–4 min long, the pre and post viewing discussion took an additional 10 min. Participants could have seen the same video more than once.

**Community-based haemoglobin testing**

We also conducted monthly anaemia testing of the women using a digital Hemocue metre. These instant results are shared at the individual, group, and village levels with the help of blood shaped cards (different colors indicating anaemia severity) and infographics appropriate for a low literacy population. Monthly community-based testing was followed by a discussion about trends in anaemia and village-level comparisons (based on the haemoglobin (Hb) readings) with neighbouring communities at both the individual and community levels. This provides ipsative feedback (information people receive about their ongoing progress over time), normative feedback (information about the particular individual's achievements relative to those of her social peer), and aspirational feedback (comparisons people make between their current state of affairs and the goals they may have set for themselves). Testing sessions lasted for an hour as community/group testing was followed up by demonstration of results and behavioural nudges for improving their haemoglobin count.

*Community Facilitators of the RANI team delivered the intervention. However, IFA tablets were provided by the community health workers. RANI started off using key influencers in the village like front line health workers to gather people for the intervention. However, as our facilitators became familiar with the community, we also accepted RANI volunteers (women from the community) who facilitated this. Some of the interventions were delivered at the household level also.

---

| Table 1 | Description of participants by study arm | | |
|---|---|---|---|
| | Treatment (n=1874) | Control (n=1867) | P value |
| Demographic | | | |
| Age (years) | 31.2 | 30.5 | 0.022 |
| Part of tribal pop | 23.6% | 31.7% | 0.267 |
| Education (years) | 6.07 | 6.17 | 0.749 |
| Breastfeeding at baseline | 21.4% | 21.7% | 0.836 |
| No of children at baseline | 1.69 | 1.68 | 0.851 |
| Exposure to non-RANI interventions | 1.2% | 2.1% | 0.283 |
| Dependent variables | | | |
| IFA use at midline | 32.0% | 3.0% | <0.001 |
| Haemoglobin at midline (g/dL) | 11.66 | 11.52 | 0.281 |
| Exposure to RANI intervention components | | | |
| No of group education sessions attended | 4.60 | 0.23 | <0.001 |
| Attended at least one group education session | 88.6% | 12.1% | <0.001 |
| No of haemoglobin tests undergone | 1.28 | 0.00 | <0.001 |
| Tested for haemoglobin at least once | 81.4% | 0.0% | <0.001 |
| No of RANI Comm videos views | 2.46 | 0.00 | <0.001 |
| Viewed at least one RANI Comm video | 79.8% | 0.2% | <0.001 |

Notes: We ran regression analyses to test the statistical significance between treatment versus control arms on demographic variables. The other non-RANI-related intervention includes anaemia-related education as one component that may have had some overlap with our intervention. We included this as a control in our models to ensure that any changes in behaviour are a result of our intervention alone.
IFA, iron folic acid; RANI, Reduction in Anaemia through Normative Innovations.

collection using proportion-to-size principles based on cluster population. From the selected home, one woman of reproductive age (between 15 and 49 years old) was chosen (randomly if more than one woman was eligible in the same home). Although our sample consisted of 3953 participants, this paper restricts the sample to those who were not pregnant at the time of the midline survey (n=3800). We do so because the primary dependent variable, taking iron and folic acid tablets, has been heavily promoted among pregnant women by the Government of India. Pregnant women are also enrolled in the health system, where physicians or community health workers provide free IFA. This is not the case among non-pregnant women, who have not been targeted as IFA recipients on the ground despite WHO and Indian government

recommendations.[24] [25] The demographic profile of participants included in our analysis is shown in table 1.

## Procedure

The baseline data were collected in September 2019 and midline data were collected 6 months later in February 2020. Local data collectors obtained informed consent from all individual participants included in the study in the local language, Odiya. Participants under the age of 18 were required to obtain the written permission of one parent or legal guardian. Data collectors orally administered a one-on-one survey to all participants, which assessed demographic information, psychosocial factors, and anaemia-related behaviours.

## Inclusion criteria

Women were eligible for inclusion in the study if they were between the ages of 15 and 49, spoke Odiya, lived in the data collection villages (either treatment or control), and did not plan to move in the next year (as this is a longitudinal study).

## Measures

### Dependent variables

The RANI study was evaluating the efficacy of a norms-based intervention to increase IFA use and reduce anaemia. Therefore, our study has two dependent variables: self-reported IFA use and objectively measured serum haemoglobin levels. We measured IFA use at midline using the interview question, 'Have you ever eaten/taken an iron tablet or syrup.' (The interviewer then held up the packet of IFA tablets for the interviewee to see). We coded this as a dichotomous variable, scored one if currently taking and 0 if not or did in the past but stopped now. We obtained haemoglobin levels from all participants at midline through point-of-care haemoglobin tests using a HemoCue photometer (in line with India's National Family Health Survey methodology). This instrument provides haemoglobin levels immediately and accurately.[26]

### Independent variables (exposure to the intervention)

We examined four independent variables. The first is a dichotomous indicator of treatment assignment that takes the value one for participants residing in intervention villages and 0 for those residing in control villages. The other three independent variables are participants' self-reports of exposure to different components of the RANI intervention. To measure *exposure* to the participatory learning modules, we took the sum of responses to six questions about how often participants had seen materials from these sessions. We used visual images from the sessions, with higher scores indicating more exposure or more frequent exposure (not seen=0, seen once or twice=1, and seen more than twice=2). Each image came from a different participatory learning module. One question asked whether or not they had participated in any of the games that were also a part of the RANI participatory group sessions (scored as no=0 and yes=1). The

total number corresponds to the number of times participants report having seen a particular image from any of the group sessions, with this being a proxy for the intensity of exposure to group participatory sessions. Less than 1% of participants (n=28) marked 'don't know' (which was coded as missing). We coded exposure to participatory learning sessions as a continuous variable (range 0–11). We assessed participants' own anaemia testing (exposure to the anaemia testing component of the intervention) with the question, 'How many times have you been tested for anaemia as part of the RANI intervention in the last 6 months?' Response options ranged from 0 (never) to 4 (more than three times). We also coded testing for anaemia as a continuous variable (range 0–4). We measured exposure to the RANI Comm videos as the sum of responses to four questions about which of the four videos they had watched. Interviewers shared an image from each video and a brief description of the story plot. Responses were treated as a continuous variable and summed across the four videos for a range from 0 to 4.

## Control variables

We asked respondents their age, highest completed level of education and whether they belonged to a scheduled tribe. IFA use at baseline was assessed exactly as described above for midline. We asked respondents about the number of children they had and whether or not they were breastfeeding. Additionally, to understand if participants had been exposed to another intervention that was not affiliated with RANI (to avoid contamination), we asked participants, 'Did you hear anything about nutrition or iron tablets from the Swabhimaan or any other program?' We coded this as a dichotomous variable (no or don't know=0 and yes=1).

## Statistical analysis

We conducted our analyses in three steps. First, we calculated frequencies and descriptive statistics of all key analytic variables by treatment and control arm and obtained p values testing the null hypothesis of no difference between the two arms via linear and logistic regression for continuous and categorical variables, respectively. These analyses covered several socioeconomic and health-related background variables; amount of exposure to each of the three intervention components between baseline and midline; and haemoglobin levels and self-reported IFA use at midline. The latter provided unadjusted intention-to-treat estimates of the overall RANI treatment effect on these primary endpoints at midline.

Second, we ran linear regressions to examine if exposure to each intervention component varied by select sociodemographic factors: age (32 years or older vs below 32 years old—the halfway point between the age range), belonging to a schedule caste or not, and education (completed primary school or not). These analyses were limited to residents of clusters assigned to the RANI intervention arm.

Third, we ran a series of regression models to examine how each intervention component individually and additively affected haemoglobin levels (linear) and IFA use (logistic) at midline and accounted for the overall effect of treatment assignment. For each outcome, model 1 includes only control variables (age, education, currently breast feeding, number of children, whether they belong to a scheduled caste or tribe, knowing anaemia status at baseline, IFA use at baseline, and whether the participant reported exposure to a non-RANI intervention), and RANI treatment assignment. The coefficients on RANI treatment assignment in these models represent adjusted intention-to-treat estimates for each outcome. In model 2, we add a set of dummy variables representing levels of exposure to the first RANI intervention component, group education sessions, with zero sessions as the reference category. The dummy variable specification provides flexibility in the nature of the relationship between number of group education sessions attended and the two outcome variables. The coefficients on the dummy variables in this model represent how midline haemoglobin and self-reported IFA use vary in relation to number of group education sessions attended. Moreover, comparison of the coefficient on the RANI treatment assignment variable in this model to the analogous coefficient in model 1 provides insight into the extent to which group education sessions contributed to the overall effect of RANI. The next two models are similar to model 2 but using haemoglobin testing (model 3) and viewing of RANI Comm videos (model 4) instead of group education session attendance. Finally, model 5 includes all three intervention components. The coefficients on the intervention component dummy variables in this model provide some insight into the unique contribution of each component, independent of the other two. And comparison of the coefficient on the RANI treatment assignment dummy variable in this model to the corresponding coefficient from model 1 represents the extent to which the three intervention components jointly contributed to the overall RANI intervention effect. We used Stata V.16 to conduct all analyses, with Huber-White clustered SEs[27] to account for the sampling and cluster randomisation design.

## RESULTS

Description of the sample included in our study is shown in panel A of table 1. Average age was 31 years old and between a quarter and a third of participants were a part of the tribal population. On average, participants completed primary school (6 years of education). Participants in both treatment and control arms had on average more than one child and fewer than two. About 21% of women in both arms were currently breastfeeding. The only statistically significant difference between the treatment and control arms was age. Women in the treatment arm were 31 years old vs 30 years old in the control arm (p<0.05). All other demographics were not statistically

**Table 2** Exposure to intervention components by age, education, and caste (treatment arm participants)

|  | Average no of group educational sessions attended | Average no of haemoglobin tests undergone | Average no pf RANI comm videos seen |
|---|---|---|---|
| **Age** | | | |
| Less than 32 years | 4.48 (SD: 2.82) | 1.33 (SD: 1.04) | 2.41 (SD: 1.55) |
| 32 years or more | 4.74 (SD: 2.95) | 1.22 (SD: 0.95) | 2.53 (SD: 1.55) |
| P value | 0.219 | 0.056 | 0.119 |
| **Education** | | | |
| Up to completed primary | 4.77 (SD: 2.88) | 1.22 (0.93) | 2.55 (SD: 1.55) |
| More than completed prim | 4.46 (SD: 2.88) | 1.32 (1.05) | 2.40 (SD: 1.55) |
| P value | 0.040 | 0.033 | 0.132 |
| **Scheduled tribe or caste** | | | |
| Yes | 4.54 (SD: 2.87) | 1.28 (SD: 1.00) | 2.44 (SD: 1.56) |
| No | 4.78 (SD: 2.90) | 1.29 (SD: 1.01) | 2.55 (SD: 1.52) |
| P value | 0.429 | 0.899 | 0.407 |

RANI, Reduction in Anaemia through Normative Innovations.

different by treatment and control. Panel B of table 1 shows that exposure to non-RANI interventions was low in both study arms.

Crude estimates of the effects of the RANI intervention at midline are shown in panel C of table 1. Participants from the treatment arm reported more IFA use at midline (32%) compared with the control arm (3%) (p≤0.001). However, haemoglobin levels were not statistically different in the two arms (11.7% vs 11.5%) (p=0.28) (see online supplemental appendix table 1). Panel D of table 1 shows that exposure to each of the intervention components was widespread among participants in the treatment arm and low among participants in the control arm. Specifically, 88% of participants in the treatment arm compared with only 12% in the control arm reported that they attended at least one group educational session (p≤0.001). Over eighty percent of participants in the treatment arm reported that they had been tested at least once as part of the intervention compared with 0% in the control arm (p≤0.001). Similarly, 80% of women in the treatment arm reported that they watched one or more RANI Comm videos, compared with less than 1% in the control arm (p≤0.001). Lastly, exposure to interventions other than RANI was minimal across both the treatment and control arms (panel B, 1.2% and 2.1%, respectively).

Table 2 shows that less educated women attended more educational sessions (4.77 compared with 4.46; p≤0.05) but more educated women had a higher average number of anaemia tests (1.32 compared with 1.22; p≤0.05). Table 2 also shows that neither age nor caste or tribal background was associated with exposure to the intervention.

As there was no crude difference between the treatment and control arms in haemoglobin level at midline (see panel C of table 1), we included the analyses linking exposure to specific intervention components to haemoglobin levels as an appendix (see online supplemental appendix table 1). However, nested logistic regression models predicting self-reported IFA use as a function of study arm and exposure to each of the intervention components are presented in table 3. As seen in model 1, after controlling for demographic variables and IFA use at baseline, simply being in the treatment arm increased a woman's odds of taking IFA by more than 16 times (OR=16.94; p≤0.001). In model 2, we added exposure to the group education sessions and found that the odds of taking IFA increased as the number of sessions attended rose. Model 3 suggests that each additional test for anaemia also showed an increase in the odds of taking IFA. Model 4 similarly shows that for each RANI Comm video that a woman watched there was an increase in her odds of taking IFA. Model 5 with all intervention components included shows that while all effect sizes drop (including the effect of RANI treatment assignment), each component still had a significant effect on IFA use. This indicates that each component explains a unique part of the variance in current use of IFA.

## DISCUSSION
In this paper, we present findings from the midline interim assessment of a multicomponent behavioural intervention intended to increase IFA use and decrease anaemia among women of reproductive age in Odisha, India. We find strong evidence that the intervention increased the prevalence of self-reported IFA use among non-pregnant women, but no evidence of an effect of the intervention on average haemoglobin levels in that group. Self-reported exposure to all three intervention components was high among women in the treatment arm, which did not vary substantially by age or membership in a scheduled caste or tribe. However, exposure to anaemia testing was higher among more educated women and exposure to group sessions was higher among less educated women. Self-reported exposure to all three

**Table 3** Adjusted ORs from logistic regression models predicting self-reported IFA use at midline as a function of treatment assignment, exposure to three intervention components, and control variables

| | Model 1 | Model 2 | Model 3 | Model 4 | Model 5 |
|---|---|---|---|---|---|
| RANI intervention overall | 16.94*** | 5.24*** | 3.62*** | 5.16*** | 2.25* |
| Group education sessions | | | | | |
| 1 (vs 0) | | 1.50 | | | 1.17 |
| 2 (vs 0) | | 2.89*** | | | 1.71* |
| 3 (vs 0) | | 2.94*** | | | 1.37 |
| 4 (vs 0) | | 3.05*** | | | 1.30 |
| 5 (vs 0) | | 3.80*** | | | 1.43 |
| 6 (vs 0) | | 5.37*** | | | 1.76* |
| 7 (vs 0) | | 5.39*** | | | 1.39 |
| 8 (vs 0) | | 8.53*** | | | 2.41* |
| 9 (vs 0) | | 8.16*** | | | 2.30** |
| 10 (vs 0) | | 9.23*** | | | 2.23* |
| 11 (vs 0) | | 11.97*** | | | 2.81** |
| Anaemia testing | | | | | |
| 1 (vs 0) | | | 4.91*** | | 3.07*** |
| 2 (vs 0) | | | 6.93*** | | 3.72*** |
| 3 (vs 0) | | | 11.94*** | | 5.65*** |
| 4 (vs 0) | | | 12.91*** | | 4.90*** |
| RANI Comm videos | | | | | |
| 1 (vs 0) | | | | 2.30*** | 1.44 |
| 2 (vs 0) | | | | 2.85*** | 1.59 |
| 3 (vs 0) | | | | 3.57*** | 1.72* |
| 4 (vs 0) | | | | 5.66*** | 2.39*** |
| Control variables | | | | | |
| Age | 0.99 | 0.99 | 1.00 | 0.99 | 0.99 |
| Education | 0.98 | 0.97 | 0.97 | 0.98 | 0.97 |
| Breast feeding | 1.35* | 1.37* | 1.41* | 1.41** | 1.43** |
| No of children | 0.99 | 0.96 | 0.96 | 0.97 | 0.95 |
| Caste/tribe | 1.31* | 1.26 | 1.31* | 1.28 | 1.26 |
| Knows anaemia status at baseline | 1.00 | 1.00 | 1.00 | 1.00 | 1.00 |
| Baseline IFA use | 2.59*** | 2.94*** | 3.30*** | 2.87*** | 3.36*** |
| Non-RANI intervention exposure | 1.85 | 1.28 | 1.57 | 1.59 | 1.36 |
| Pseudo R2 | 0.20 | 0.24 | 0.24 | 0.23 | 0.26 |

*P<0.05, **p<0.01, ***p<0.001.
IFA, iron folic acid; RANI, Reduction in Anaemia through Normative Innovations.

intervention components was positively associated with IFA use, and these exposures accounted for most but not all the overall effect of the RANI intervention on IFA use among non-pregnant women.

Given that we saw very little difference in exposure to the intervention components by age, education or caste, it appears that the RANI Project's outreach efforts did not disproportionately favour those with higher privilege in society. Indeed, even though exposure to the intervention did not differ by tribal status or age, in our multivariate models we found some evidence that those with lesser education had higher IFA uptake. We view this finding rather optimistically, given that interventions have the potential for exacerbating existing differences in society along access and socioeconomic lines,[28] as has been documented by the literature on the knowledge gap hypothesis.[29 30]

We saw strong evidence that testing women for anaemia (through finger-prick haemoglobin counts) was associated with IFA uptake: those who got tested more often

were more likely to consume IFA tablets. The reasons for this are not known precisely, but we do have a few explanations. First, it could be that testing revealed to women the low haemoglobin counts, which spurred them to act to improve their scores. Our formative assessment had revealed that feeling weak was often a part of women's self-identity, with women often believing that fatigue was par for the course[19] and that this was part of the gendered norms for women.[20] We suspect that, in contrast, an objective measure of iron in the blood, observed through haemoglobin testing, quantified and thus provided precision to a phenomenon woman had come to accept as a vague notion of fatigue. We found that testing was highly demanded and motivational. Indeed, our process evaluation data indicate that, despite a high (60%) coverage rate of anaemia testing for women in our intervention catchment areas, we have been unable to meet the demand for testing and repeat testing. If we were able to meet the demand, the impact of testing may have been even higher.

A second reason why testing may have motivated women to consume IFA tablets pertains to observational learning around social norms. Due to test results being displayed publicly in the community, and most educational sessions linked the test results with IFA consumption, women likely came to link their consumption behaviours with haemoglobin readings—not only for themselves but also for what they had observed among others in their community, where haemoglobin testing was a monthly event. These events were so popular that, in some villages, the waiting list for testing ran up to 90 names (we were unable to accommodate more than 15 women per village per month, given the study design and resources). This public display of IFA testing and haemoglobin results also normalised women prioritising their own health potentially shifting gender norms that solely focus on pregnant women's health or the health of the family. Although these underlying reasons are somewhat speculative, they do point to the need to study more precisely the link between testing and IFA consumption.

Each educational participatory learning session that women attended significantly increased IFA use. These sessions were designed primarily to raise awareness and improve knowledge about anaemia, iron-rich foods, and diet diversity. We also included discussions about gender roles, how eating order in the home disfavours women's health, and the need to remain strong to be able to take care of others. We have few reasons to believe that knowledge in these domains would directly translate into behaviours. However, a significant body of work demonstrates that knowledge about health is a necessary, though not sufficient, condition for behaviour change[31 32] and that people often hold knowledge in abeyance, to be acted on at the appropriate time.[33] From this perspective, it seems that educational sessions may have played an important role.

Finally, we found that exposure to the communication videos was also significantly associated with increased IFA

uptake. We suspect that the underlying reason for this finding pertains to another normative component of the intervention. The four videos we developed targeted different audiences, including adolescents, husbands, mothers-in-law, and women of reproductive age. As an explicit campaign strategy, each video was shown to each of the parties, including those who were not the explicit target audience for the particular video. This was done so that each group came to understand that the other groups were also being targeted by the intervention. So, for example, adolescent girls saw that there were videos that also addressed men and older women. Similarly, men saw that the videos targeted other men and other women. The overall strategy was to communicate the message that the entire community was engaged in the task of reducing anaemia.

In this paper, we assessed how each of the intervention components affected IFA use. This helps elucidate where this intervention should focus for the remaining trial, provides a clear picture of dose response, and highlights where other social norms programme implementors may want to focus their efforts. In another midline paper, we assessed the effect of social norms on IFA use and found that changes in descriptive and collective norms (but not injunctive norms) were associated with changes in self-reported IFA use.[34]

It is also important to discuss the lack of an effect of the RANI intervention on haemoglobin use despite an increase in self-reported IFA use. One plausible explanation of these disparate findings is that the increase in IFA use among members of the treatment arm is real but is not sufficient in magnitude or duration to produce a corresponding increase in haemoglobin levels. Alternatively, it could be attributable to differential misreporting of IFA use among treatment group members, a form of courtesy or social desirability bias. Findings from the end line data collection, planned to take place at the end of the project, may help to adjudicate between these two explanations.

Of course, our study is not without limitations. First and foremost, while treatment assignment is objectively measured on the basis of project administrative data, exposure to the three intervention components is self-reported and may be subject to some level of misreporting. Furthermore, it is possible that women who participate in the intervention may be more motivated to change behaviour in general and are already more inclined to take IFA. Therefore, table 3 could overstate the effects on IFA use of exposure to the intervention components, as well as the contributions of those components to the overall RANI effect. However, IFA use is still a result of participating in the RANI intervention, as participation led to IFA use so we may have simply captured women who were farther along in their readiness to change. Additionally, although our study used a representative sample of areas and women in our focal areas, findings may not be generalisable outside of that setting.

Despite these limitations, our study has strengths: the intervention is based on formative research; it is co-designed by stakeholders from the area; we use a theoretically driven and adaptive approach; and the study design itself is robust, with an underlying randomisation to experimental arms.

## Conclusion

Our findings show that a social norms-based intervention can be successful in increasing IFA use. They also demonstrate that unique intervention components separately and altogether impact this success. All three components appear to contribute something to the overall effect of the intervention. While all three intervention components tapped into social norms messaging, haemoglobin testing provided individual health information, village level health information, a comparison to other villages, and changes in health information over time. This multilevel component, coupled with the other two components, may help women reach the tipping point to take and adhere to IFA. While IFA use shows promise, haemoglobin levels may need more time to show significant changes, especially among non-pregnant women who were not taking IFA and who only take it weekly. End line results will elucidate more information about the full RANI intervention effects.

**Author affiliations**
[1]Institute for Health and Aging, Department of Social and Behavioral Sciences, University of California, San Francisco, California, USA
[2]Department of Prevention and Community Health, George Washington University, Milken Institute School of Public Health, Washington, District of Columbia, USA
[3]Deptartment of Health Science, Boston University Sargent College, Boston, Massachusetts, USA
[4]IPE Global Limited, Delhi, India
[5]DCOR Consulting Pvt. Ltd, Bhubaneswar, Odisha, India
[6]Department of Biomedical Laboratory Sciences, George Washington University, School of Medicine and Health Sciences, Ashburn, Virginia, USA
[7]Department of Health, Behavior & Society, Johns Hopkins University Bloomberg School of Public Health, Baltimore, Maryland, USA

**Contributors** ES: conceptualisation, methodology, analysis, writing—original draft. Responsible for the overall content as the guarantor. IP: conceptualisation, methodology, writing—review and editing. JB: methodology, analysis, writing—review and editing. HY: methodology, data curation, writing—review and editing. LP, SM: project administration, supervision, writing—review and editing. RG: writing—review and editing. RR: conceptualisation, methodology, supervision, funding acquisition, writing—review and editing.

**Funding** This work was supported by a grant from the Bill & Melinda Gates Foundation (OPP1182519) to the George Washington University, RR, principal investigator.

**Disclaimer** The funders had no role in study design, data collection and analysis, decision to publish, or preparation of the manuscript.

**Competing interests** None declared.

**Patient and public involvement** Patients and/or the public were involved in the design, or conduct, or reporting, or dissemination plans of this research. Refer to the Methods section for further details.

**Patient consent for publication** Not applicable.

**Ethics approval** This study was approved by the George Washington University Institutional Review Board (FWA00005945), Sigma Science and Research, an independent IRB located in New Delhi, India, and the Indian Council for Medical Research's Health Ministry's Screening Committee. Participants gave informed consent to participate in the study before taking part.

**Provenance and peer review** Not commissioned; externally peer reviewed.

**Data availability statement** Data are available in a public, open access repository. Our data can be found in the GWU Scholar Space online repository and by request from the first author. https://scholarspace.library.gwu.edu/.

**ORCID iD**
Erica Sedlander http://orcid.org/0000-0002-5128-669X

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
