## [Reviewer comments · BMJ Open]

ARTICLE DETAILS

TITLE (PROVISIONAL)	How does a social norms-based intervention affect behavior change? Interim Findings from a cluster randomized controlled trial in Odisha, India
AUTHORS	Sedlander, Erica; Pant, Ichhya; Bingenheimer, Jeffrey; Yilma, Hagere; Patro, Lipika; Mohanty, Satya; Ganjoo, Rohini; Rimal, Rajiv

VERSION 1 – REVIEW

REVIEWER	Sarah Cotterill University of Manchester, Centre for Biostatistics
REVIEW RETURNED	14-Jun-2021

GENERAL COMMENTS	Design The paper reports interim results from an observational study within a cluster randomised controlled trial in India, where 3953 women from 15 clusters of villages (81 villages in total) were randomised to a social norms intervention (RANI) or usual care. Women in the intervention group were offered a complex package of interventions with the aim of raising the consumption of iron supplements. The complex intervention included 3 components, all based on social norms: group education sessions, anaemia testing, communication videos. Women in the control group were offered usual care. The primary outcome in the trial protocol is anaemia (haemoglobin levels using a point-of-care test) and there were a number of secondary outcomes. This paper is not the main trial results paper, and I assume the authors intend to produce the traditional ITT results at a later time. Rather, the purpose of this paper is ‘to examine the extent to which each intervention component contributed to the overall effects of the intervention. In addition, because the influence of the intervention may be different across subgroups, we also examine how susceptibility to intervention impact varied by age, caste, education, and communication activity’ in a sub-group of the trial population, i.e. women who were not pregnant (3800 out of 3953 women). The paper reports two outcome measures: haemoglobin levels and self-reported consumption of folic acid. There is no explanation of why these two were chosen from among the cluster of measures that were collected, and I would like to an explanation for this decision added to the paper. The ‘study design’ and ‘participants’, ‘procedure’, ‘inclusion criteria, sections of the paper describe the host trial with brief reference to this study. I would suggest addition of new separate section describing the host trial, separating that out from the design of this study. It would be helpful to explain the timing of the midline or midterm assessment (are these both the same assessment, or two different ones?) in relation to the overall plan of assessment. The CONSORT checklist for RCTS does not seem the appropriate
---

	reporting checklist to use for this study, which is an observational analysis of the trial data. One of the observational tools would be more appropriate: STROBE or one of the extensions. Results In an ITT analysis there was no difference in the primary outcome (haemoglobin levels were 11% vs 11.5%). There was a difference between the groups in self-reported consumption of iron folic acid (treatment 31.6% vs control 3%). Self-reported consumption of iron folic acid was not one of the outcomes listed in the protocol. The subsequent analyses treat the trial data as an observational study, comparing across sub-groups of the participants. Specifically, the authors compare across sub-groups of women who reported that they had taken part in three components of the intervention (group education sessions, anaemia testing, communication videos). Firstly (table 3), the authors report exposure to the three components of the intervention by demographics, using linear regression, and find that the only difference between those who took up the components and those who did not was age, with older people (32 years and over) being more likely to take part. It is always better to avoid categorising a continuous variable in a regression analysis, and there are dangers in choosing optimal cut-points.¹ Reassuringly, neither education nor tribal membership was associated with uptake of the intervention. Secondly (table 4), the authors present logistic regression models examining the relationship between self-reported iron and folic acid consumption (outcome) and the three intervention components, controlling for demographic variables, baseline iron folic acid consumption and group allocation. The findings from this are that women who took up the offer of all three components that encouraged consumption of iron folic acid (over 80% of the intervention group and a small % of the control group) were more likely than other women to self-report that they consumed iron folic acid, and that this effect was seen separately for each of the three components. The authors conclude that each component of the intervention contributes to variance in self-reported use of iron folic acid. Discussion There are two strong limitations, which the authors acknowledge in the discussion section. Firstly, the difference in effect between the two outcomes may suggest either that exposure to the intervention led to higher iron consumption and that has not yet had time to influence haemoglobin level, or that the intervention encouraged women to over-report their consumption of iron. Secondly, exposure to the interventions is by self-report, rather than objectively measured. Another limitation could be added: there will plausibly be a relationship between the two: women who comply with the intervention components may also be more compliant with taking iron supplements. While it is tempting to compare the results by treatment actually received, the danger is that the comparison is then between a compliant group and a non-compliant group, and compliance is often related to the outcome: 'Results of analysis by treatment actually received may suffer from a bias introduced by using compliance, a factor often related to outcome independently of the treatment received, to determine the groups for comparison. The extent and nature of this bias will be related to the definition of compliance in an as treated analysis, a definition which could be unintentionally self-serving'.² Several ways of considering compliance in trials have been put forward, including instrumental variable (IV) and complier average
--	--

	casual effect (CACE) analyses.³ In this case, the authors are not considering compliance to the intervention overall, but rather to the individual components. However, I think it would be helpful for them to reflect on this literature in the discussion and consider the implications for their findings. References  1. Sauerbrei W, Bland M, Evans SJW, Riley RD, Royston P, Schumacher M, et al. Doug Altman: Driving critical appraisal and improvements in the quality of methodological and medical research. Biometrical Journal 2021;63:226-46. https://doi.org/https://doi.org/10.1002/bimj.202000053 2. Lee YJ, Ellenberg JH, Hirtz DG, Nelson KB. Analysis of clinical trials by treatment actually received: is it really an option? Statistics in medicine 1991;10:1595-605. https://doi.org/10.1002/sim.4780101011 3. Ye C, Beyene J, Browne G, Thabane L. Estimating treatment effects in randomised controlled trials with non-compliance: a simulation study. BMJ Open 2014;4:e005362. https://doi.org/10.1136/bmjopen-2014-005362
--	---

REVIEWER	Uttara Partap Harvard University T H Chan School of Public Health, Global Health and Population
REVIEW RETURNED	17-Jan-2022

GENERAL COMMENTS	This manuscript describes the baseline and midline assessment of a cluster-randomized controlled trial which aims to increase IFA intake among women of reproductive age through a social norms-based intervention. This is a very interesting and relevant study, especially given its population, and intervention. The manuscript is generally well-written and it is clear that much work has gone into designing and implementing the study. I do have some comments and questions for the authors to consider, mainly to ensure that analyses especially are rationalized and presented clearly - below. Major comments  1. Methods: Although Table 1 gives a brief summary of each component of the intervention, the following additions would be helpful information, especially to contextualize how the authors defined their variables for exposure to each component:  a. What was the total duration of the components? (given midline is 6mo, I understand 12mo but good to be explicit) b. For video viewing, was this monthly also (not specified in Table 1)? c. How were people invited to take part in these interventions? d. Who delivered each intervention? e. For video viewing – were all four videos shown in one sitting with a group, or was the idea to show different videos at different times? f. For video viewing – could it be that one person saw the same video multiple times? I think elements of these are in the study protocol, but it would be good to reiterate these specific points very briefly where appropriate in this paper. 2. Methods/Main analysis (Table 4):  a. It is not entirely clear what exact exposure variables were used for the components – could the authors reiterate in the footnotes (and perhaps clarify as relevant in the Methods)?
---

	i. Exposure to participatory learning modules: was the final variable actually the number of modules the participant attended? The Methods are unclear to me, and seemed to indicate instead that there was an overall categorical variable about frequency of exposure (if so, it should be included as such in the model). ii. Own anaemia testing – seems that this is a categorical variable with options 0=never to 4=more than 3 times – again in this case this should technically be included as a categorical rather than continuous variable. iii. In Table 4, have the authors considered testing the linearity of effect for continuous variables such as video watching (e.g. is increase in odds actually the same across categories of video watching from 1 to 4?) This could be checked using likelihood ratio tests. 3. From the Discussion, page 18 (of the proof) first para, the authors note that testing was so popular that there was often a waiting list. Could there have been a possibility that a study participant wanted to get tested but was unable to do so, and if so, did the authors collect data on how many times they had the intention of getting tested? Result suggest a pretty strong association already, but I wonder whether the effect of this component may be different (perhaps greater?) if intention to test is taken into account. 4. Table 3: a. Could the authors please clarify (maybe in a footnote) the exact analyses undertaken to arrive at specifically the P values? I am assuming that the P values are from Student's t-tests/Chi sq tests, but equally they could be from regressions as described in page 12 (of proof) para 2 "Second, we ran linear regressions to examine if". b. Again related to above: "Proportion undergoing at least one haemoglobin test" – I am assuming that this was based on a binary variable technically, so if P values came from a linear regression, I would recommend that the authors use logistic/Poisson for this (or if a crude method was used, it would make sense to check that this was Chi squared test or similar). Also here the number in each category versus overall N would be extremely informative. c. Would it be possible to also include the range of number of group educational sessions attended (if this information was collected) and number of RANI Comms videos viewed? (I suppose for RANI Comms videos this is 1-4?). 5. Main analysis (Table 4): I understand the motivation for including assignment to RANI intervention overall as an explanatory variable in models, and it is interesting to see a strong effect of this independent of the other variables. I do wonder whether including both the overall intervention and additionally the elements that comprise the overall intervention into a singular model may actually be over-adjusting in a sense, given that the aim is to examine the effect of the singular components (i.e. we see less of an effect of the individual components because assignment to overall intervention is accounting for some of this). Could the authors comment on this – have they considered use of a model that does not include overall intervention? 6. Results (page 14 of proof, para 3): It appears the authors decided not to undertake regression analyses for haemoglobin as an outcome on the basis of no effect in crude analyses. Especially if such analyses were pre-planned, it would be worth undertaking these – especially given that regression models allow for the accounting of potential confounding (which could be one factor leading to a suggestion of "no effect" in crude analyses). It would be good to see this, even if only in supplementary tables.
--	---

	7. Communicating about the intervention – a. Did the authors also do linear regressions with this variable also (“Second, we ran linear regressions ...” page 12 of proof, para 2) and could the authors also summarise in Table 3 the outcomes by this variable as it seems Table 3 focuses on the named effect modifiers? b. This is not mentioned at all in Results – could the authors add wording around this? Minor comments 8. Abstract: the target population of the trial (specifically: women of reproductive age) is not mentioned. It would be good to make this clear at the outset. 9. Under strengths and limitations of this study, the authors mention that this is a “large double blinded cluster randomized trial”. I am not entirely sure I follow this. I understand from the protocol that most study staff including investigators and data collectors are blinded to treatment allocation. However, I am not sure that participants can be blinded to which arm they are in, effectively meaning that this is not strictly double blinded. Could the authors please justify more clearly or amend the use of this term? 10. Page 7 (of the proof) para 3 under “Intervention Development”, the sentence “To explore women’s social norms within focus groups, we used [18].” this seems to end prematurely – or perhaps the authors intended to refer to reference 18. In any case it would be helpful to actually spell this out (I understand this is the use of vignettes?) for ease of reading. 11. Table 2: I am unsure as to how t-tests were used for categorical variables like exposure to a non-RANI intervention. The methods suggest that a Chi2 test was used (which is more appropriate and makes sense) – could the authors please clarify in the table? 12. Results last para last sentence, I think the authors mean to refer to Table 4. 13. Discussion – strengths and limitations – absence of effect of intervention on haemoglobin use is not a limitation, just a finding. 14. In certain cases, a reference is “Anonymous” (references 8, 19, 20, 21) – do these relate to participant views, or if not, could the authors check any errors in referencing? (reference 21 for example to the RANI project seems to go to “Anonymous 2020” whereas I would assume this would go to the protocol paper)
--	---

VERSION 1 – AUTHOR RESPONSE

Reviewer Comment	Response
Design: The paper reports two outcome measures: haemoglobin levels and self-reported consumption of folic acid. There is no explanation of why these two were chosen from among the cluster of measures that were collected, and I would like to an explanation for this decision added to the paper.	Thank you for suggesting that we clarify this. We added a sentence: “The RANI study is evaluating the efficacy of a norms-based intervention to increase IFA use and reduce anemia. Therefore, our study has two dependent variables: self-reported IFA use and objectively measured serum hemoglobin

	levels.”
The ‘study design’ and ‘participants’, ‘procedure’, ‘inclusion criteria’, sections of the paper describe the host trial with brief reference to this study. I would suggest addition of new separate section describing the host trial, separating that out from the design of this study. It would be helpful to explain the timing of the midline or midterm assessment (are these both the same assessment, or two different ones?) in relation to the overall plan of assessment.	Thank you for allowing us to clarify. This study is very much the host trial. We are simply reporting interim midterm findings from the main trial. We added some text including timing to the abstract and methods clarify this.
The CONSORT checklist for RCTS does not seem the appropriate reporting checklist to use for this study, which is an observational analysis of the trial data. One of the observational tools would be more appropriate: STROBE or one of the extensions.	Given that this is an interim analysis of the main trial which is a randomized controlled trial, we believe that the CONSORT checklist is appropriate.
In an ITT analysis there was no difference in the primary outcome (haemoglobin levels were 11% vs 11.5%). There was a difference between the groups in self-reported consumption of iron folic acid (treatment 31.6% vs control 3%). Self-reported consumption of iron folic acid was not one of the outcomes listed in the protocol.	In the RCT study protocol, we state, “H1. Changes in women from baseline to end line in the intervention arm will be significantly greater than corresponding changes in the control arm in the following outcomes: (a) anemia status, (b) IFA use.”
The subsequent analyses treat the trial data as an observational study, comparing across sub-groups of the participants. Specifically, the authors compare across sub-groups of women who reported that they had taken part in three components of the intervention (group education sessions, anaemia testing, communication videos).	We responded to this comment in the first two paragraphs (above the table) in the response to reviewers.
Firstly (table 3), the authors report exposure to the three components of the intervention by demographics, using linear regression, and find that the only difference between those who took up the components and those who did not was age, with older people (32 years and over) being more likely to take part. It is always better to avoid categorising a continuous variable in a regression analysis, and there are dangers in choosing optimal cut-points. ¹ Reassuringly, neither education nor tribal membership was associated with uptake of the intervention.	We agree with the reviewer that dichotomizing a continuous variable (in this case, age) can be problematic because it throws out true variation. We did two things to address this. First, we ran regression models with each intervention component as the dependent variable and age in years as the

	independent variable. The coefficients on age in these three models were 0.017 (p=0.023) for number of educational sessions attended, 0.002 (p=0.332) for number of hemoglobin tests undergone, and 0.010 (p=0.001) for RANI Comm videos seen. These are consistent with the findings using dichotomized age reported in Table 3. Second, we analyzed all three outcomes using a seven-level age category variable (15-19, 20-24, ..., 45-49) because the specification with continuous age could also be problematic if the relationship between age and uptake of each of the three intervention components is curvilinear. Across these three models, all age groups had average levels of intervention component exposure than the youngest group (15-19 year olds), but only some of these elevated exposure levels were statistically significantly different from zero (the 25-29, 35-39, and 40-44 year old groups for education sessions; the 20-24 and 35-39 year old groups for anemia testing; and the 35-39 and 40-44 year old groups for RANI Comm videos). Because the results of these more sophisticated analyses largely support the easier-to-present results of the analyses using
--	---

	dichotomized age, we retain the original analyses in the manuscript and simply mention briefly that these other analyses produce similar results.
Discussion: Another limitation could be added: there will plausibly be a relationship between the two: women who comply with the intervention components may also be more compliant with taking iron supplements.	Thank you for drawing this to our attention. We agree with the reviewer that some of the variation in exposure to the intervention components is endogenous, i.e., a result of self-selection rather than treatment assignment. This in turn implies that the associations between exposure to those intervention components and our two outcome variables (self-reported IFA use and actual anemia levels at midline) could be spurious. We attempted to address this in the analysis by controlling for key baseline variables including age, parity, IFA use at baseline, hemoglobin at baseline, and other variables. Nevertheless, the possibility of omitted variable bias remains. We have addressed this limitation more thoughtfully in the discussion section of the manuscript.
While it is tempting to compare the results by treatment actually received, the danger is that the comparison is then between a compliant group and a non-compliant group, and compliance is often related to the outcome: 'Results of analysis by treatment actually received may suffer from a bias introduced by using compliance, a factor often related to outcome independently of the treatment received, to determine the groups for comparison. The extent and nature of this bias will be related to the definition of compliance in an as treated analysis, a definition which could be unintentionally self-serving'.²	As noted, we agree with the reviewer's point that the associations between our three intervention exposure variables and midline IFA use and hemoglobin levels could be partially spurious, in spite of our controlling for

Several ways of considering compliance in trials have been put forward, including instrumental variable (IV) and complier average casual effect (CACE) analyses.³ In this case, the authors are not considering compliance to the intervention overall, but rather to the individual components. However, I think it would be helpful for them to reflect on this literature in the discussion and consider the implications for their findings. References  1. Sauerbrei W, Bland M, Evans SJW, Riley RD, Royston P, Schumacher M, et al. Doug Altman: Driving critical appraisal and improvements in the quality of methodological and medical research. Biometrical Journal 2021;63:226-46. https://doi.org/https://doi.org/10.1002/bimj.202000053 2. Lee YJ, Ellenberg JH, Hirtz DG, Nelson KB. Analysis of clinical trials by treatment actually received: is it really an option? Statistics in medicine 1991;10:1595-605. https://doi.org/10.1002/sim.4780101011 3. Ye C, Beyene J, Browne G, Thabane L. Estimating treatment effects in randomised controlled trials with non-compliance: a simulation study. BMJ Open 2014;4:e005362. https://doi.org/10.1136/bmjopen-2014-005362 r. Uttara Partap, Harvard University T H Chan School of Public Health	several baseline variable. Unfortunately we do not believe that an instrumental variables or complier average causal effects (CACE) analysis is possible here because we have only one exogenous variable upon which to instrument: treatment assignment. Put another way, for each of our intervention exposure variables, the other two intervention exposure variables represent violations of the exclusion restriction. Our understanding is that we would need at least three exogenous instrumental variables for a situation like this where there are three forms of “compliance.” We do appreciate the suggestion and accompanying references.
Major comments  1. Methods: Although Table 1 gives a brief summary of each component of the intervention, the following additions would be helpful information, especially to contextualize how the authors defined their variables for exposure to each component:  a. What was the total duration of the components? (given midline is 6mo, I understand 12mo but good to be explicit) b. For video viewing, was this monthly also (not specified in Table 1)? c. How were people invited to take part in these interventions? d. Who delivered each intervention? e. For video viewing – were all four videos shown in one sitting with a group, or was the idea to show different videos at different times? f. For video viewing – could it be that one person saw the same video multiple times? I think elements of these are in the study protocol, but it would be good to reiterate these specific points very briefly where appropriate in this paper. 	We’ve responded to all of the comments in Table 1.
 2. Methods/Main analysis (Table 4):  a. It is not entirely clear what exact exposure variables were used for the components – could the authors reiterate in the footnotes (and perhaps clarify as relevant in the Methods)? 	A. We’ve clarified this in the methods section under “independent variables (exposure to the intervention).”
 i. Exposure to participatory learning modules: was the final variable 	We’ve described in more

actually the number of modules the participant attended? The Methods are unclear to me, and seemed to indicate instead that there was an overall categorical variable about frequency of exposure (if so, it should be included as such in the model).	detail how we measured exposure to the participatory learning modules.
ii. Own anaemia testing – seems that this is a categorical variable with options 0=never to 4=more than 3 times – again in this case this should technically be included as a categorical rather than continuous variable.	The reviewer is correct that own anemia testing is an ordinal variable coded 0 (for never) to 4 (four or more times). While treating it as a continuous independent variable, as we did in the original version of the manuscript, may be defensible if the relationship is in fact close to linear, we have replaced the original model in Table 4 with one in which anemia testing is represented by a set of four dummy variables comparing women who had one, two, three, and four or more tests to those who had zero tests. This dummy variable specification results in a greater reduction in the coefficient on RANI treatment assignment. In the original version that coefficient was reduced from 16.9 to 8.1 (Model 1 to Model 3). With the dummy variable specification of Model 3 it is reduced even further, to 3.6. This suggests to us that, as the reviewer suggests, the dummy variable specification is superior in that it more fully captures the form of the relationship between anemia testing and self-reported IFA use.
iii. In Table 4, have the authors considered testing the linearity of effect for continuous variables such as video watching (e.g. is increase in odds actually the same across categories of video watching from 1 to 4?) This	We thank the reviewer for this useful suggestion. As already noted, we replaced the original

could be checked using likelihood ratio tests.	version of Model 3 in Table 4 with a model using a dummy variable specification for anemia testing. We did the same for the other two intervention components – number of RANI group education sessions attended, and number of RANI Comm videos viewed. Now, Models 2, 3, 4, and 5 all use dummy variable specifications for the intervention component exposure variables.
3. From the Discussion, page 18 (of the proof) first para, the authors note that testing was so popular that there was often a waiting list. Could there have been a possibility that a study participant wanted to get tested but was unable to do so, and if so, did the authors collect data on how many times they had the intention of getting tested? Result suggest a pretty strong association already, but I wonder whether the effect of this component may be different (perhaps greater?) if intention to test is taken into account.	Yes, there was a waiting list so not everyone was able to get tested as often as they would have liked but everyone who was in the intervention arm (and wanted to) was able to get tested at least twice throughout the course of the intervention. We tested between 15-50 women per village per month depending on size of the village and if we had to, we prioritized women who had previously tested with moderate or severe anemia rather than those with mild or no anemia. Unfortunately, we did not measure intention to test as this was done every month but we did add to the discussion section stating that if we had met demand, the effect may have been even higher.
4. Table 3: a. Could the authors please clarify (maybe in a footnote) the exact analyses undertaken to arrive at specifically the P values? I am assuming that the P values are from Student's t-tests/Chi sq tests, but equally they could be from regressions as described in page 12 (of proof) para 2	A. We have rewritten the text of the Statistical Analysis section of the manuscript to clarify the what tests used to obtain

“Second, we ran linear regressions to examine if”.

b. Again related to above: “Proportion undergoing at least one haemoglobin test” – I am assuming that this was based on a binary variable technically, so if P values came from a linear regression, I would recommend that the authors use logistic/Poisson for this (or if a crude method was used, it would make sense to check that this was Chi squared test or similar). Also here the number in each category versus overall N would be extremely informative.

c. Would it be possible to also include the range of number of group educational sessions attended (if this information was collected) and number of RANI Comms videos viewed? (I suppose for RANI Comms videos this is 1-4?).

the all p-values in Tables 2, 3, and 4. All of those p-values are based on results of linear (for dichotomous dependent variables) and linear (for continuous dependent variables) regression analyses. We prefer these because more conventional approaches using Pearson’s chi-square and independent samples t-tests are not easily adapted to the non-independence that arises because of the nesting of respondents within villages/clusters. In the regression approach this clustering is easily handled using robust (clustered) standard errors (i.e., the `vce(cluster varname)` option in Stata’s `regress` and `logistic` commands).

B. Thank you for this suggestion. We referenced the changes to how we measure hemoglobin above stating that we’ve made it continuous with dummy variables.

C. We measured exposure to images that were shown during participatory sessions as a proxy for participation in each session. Therefore, we cannot report on number of sessions attended. We only report whether they report that

	they have seen these images or not (range 0-11). Correct -we reported the range of RANI comm videos as 0 to 4 in the methods section. Interviewers shared an image from each video and a brief description of the story plot.
5. Main analysis (Table 4): I understand the motivation for including assignment to RANI intervention overall as an explanatory variable in models, and it is interesting to see a strong effect of this independent of the other variables. I do wonder whether including both the overall intervention and additionally the elements that comprise the overall intervention into a singular model may actually be over-adjusting in a sense, given that the aim is to examine the effect of the singular components (i.e. we see less of an effect of the individual components because assignment to overall intervention is accounting for some of this). Could the authors comment on this – have they considered use of a model that does not include overall intervention?	We believe it is necessary to include RANI treatment assignment as a variable in the regression models in Table 4. This is because one of our main substantive questions is about the extent to which the effect of the RANI intervention overall is attributable to uptake of each of the three different intervention components. We gauge that by comparing the coefficient on treatment assignment in models 2 through 5, to that in model 1. If we did not the treatment assignment variables in all five models, we would not be able to make these comparisons.
6. Results (page 14 of proof, para 3): It appears the authors decided not to undertake regression analyses for haemoglobin as an outcome on the basis of no effect in crude analyses. Especially if such analyses were pre-planned, it would be worth undertaking these – especially given that regression models allow for the accounting of potential confounding (which could be one factor leading to a suggestion of “no effect” in crude analyses). It would be good to see this, even if only in supplementary tables.	The reviewer is correct that we previously decided not to proceed with analyses like those presented in Table 4 but with hemoglobin at midline rather than self-reported IFA use at midline as the endpoint. And the reviewer is also correct about our reason for this: the lack of a main, intention-to-treat effect on midline hemoglobin of RANI treatment assignment. We agree, however, that

	doing such an analysis makes sense from the point of view of completeness. We have done so and the result is presented in Supplementary Table 1.
7. Communicating about the intervention – a. Did the authors also do linear regressions with this variable also (“Second, we ran linear regressions ...” page 12 of proof, para 2) and could the authors also summarise in Table 3 the outcomes by this variable as it seems Table 3 focuses on the named effect modifiers? b. This is not mentioned at all in Results – could the authors add wording around this?	Given that none of the interactions were significant and we did not make any apriori hypotheses about them, we agree that we either needed to add more detail about them in the methods, results, and discussion or remove them from the paper altogether. Since they did not add anything to the overall story, and our paper is already quite long, we decided to remove them.
Minor comments 8. Abstract: the target population of the trial (specifically: women of reproductive age) is not mentioned. It would be good to make this clear at the outset.	We added this to the abstract
9 Under strengths and limitations of this study, the authors mention that this is a “large double blinded cluster randomized trial”. I am not entirely sure I follow this. I understand from the protocol that most study staff including investigators and data collectors are blinded to treatment allocation. However, I am not sure that participants can be blinded to which arm they are in, effectively meaning that this is not strictly double blinded. Could the authors please justify more clearly or amend the use of this term?	We removed the word “double” from the strengths section
10. Page 7 (of the proof) para 3 under “Intervention Development”, the sentence “To explore women’s social norms within focus groups, we used [18].” this seems to end prematurely – or perhaps the authors intended to refer to reference 18. In any case it would be helpful to actually spell this out (I understand this is the use of vignettes?) for ease of reading.	Thank you for pointing that out. We completed the sentence. It must have been mistakenly cut.
11. Table 2: I am unsure as to how t-tests were used for categorical variables like exposure to a non-RANI intervention. The methods suggest	The p-values in Table 2 are based on linear (for

that a Chi2 test was used (which is more appropriate and makes sense) – could the authors please clarify in the table?	continuous variables) and logistic (for dichotomous ones) regression models in which RANI treatment assignment was the sole independent variable. As noted above, we used regression models to compare the groups because of the ease of accounting for the nesting of respondents within clusters using the vce(cluster varname) option in Stata’s regress and logistic commands. We have clarified this in the test of the Statistical Analysis section.
12. Results last para last sentence, I think the authors mean to refer to Table 4.	Thank you for spotting that – corrected.
13. Discussion – strengths and limitations – absence of effect of intervention on haemoglobin use is not a limitation, just a finding.	We moved this out of the limitations section.
14. In certain cases, a reference is “Anonymous” (references 8, 19, 20, 21) – do these relate to participant views, or if not, could the authors check any errors in referencing? (reference 21 for example to the RANI project seems to go to “Anonymous 2020” whereas I would assume this would go to the protocol paper)	We kept references to the study protocol and formative research of the RANI project anonymous to maintain blind review, but we added them in for the revision as requested by the Editor including the protocol paper.
-Please revise the title of your manuscript to include the research question, study design and setting. This is the preferred format of the journal.	Revised.
-In your abstract and throughout, please describe your study more clearly as a (secondary) analysis of data from a clinical trial.	This is not a secondary analysis of data. This is an interim analysis of the main trial. We are simply stating interim results from the baseline and midline. We’ve made this clear in multiple places

	throughout the manuscript.
-Please ensure that all competing interests for authors are declared, including paid employment with companies.	Added.
-Please ensure that your abstract is formatted according to our Instructions for Authors: http://bmjopen.bmj.com/pages/authors/#research .	Revised.
Please include the trial recruitment dates and trial registration number.	Trial dates are listed in the methods section and trial registration is listed at the end of the paper: Trial Registration This trial was registered with Clinical Trial Registry- India (CTRI) (CTRI/2018/10/016186) on 29 October 2018.
Please include the start and end dates for the study in the Methods section.	Included.
-Please remove the protocol file from your submission, and instead cite your published protocol in the Methods section.	Cited.
Please update references 8 and 19-21 - these currently do not show author(s) or titles.	We kept references to the study protocol and formative research of the RANI project anonymous to maintain blind review, but we added them in for the revision.
We noted that there are some published papers reporting early analyses from this project in the literature already, but which are not cited/discussed in the paper that we could see. For example readers might note that there is a paper in WHO Bulletin that seems to report midline data and some of the outcomes reported here overlap/are similar to those reported in the present paper: https://www.ncbi.nlm.nih.gov/pmc/articles/PMC8542261/	We've discussed the findings for both papers and how they're unique from the findings in this paper in this version. As previously noted, we also de-anonymized all papers that come from this

- We also noted https://journals.plos.org/plosone/article?id=10.1371/journal.pone.0249646 although this seems to have a somewhat different focus. - It would be good if the authors could cite those other papers, make it clearer what outcomes/analyses from the trial have already appeared in the literature and clarify the distinctions between the present report and those already published, making sure readers understand the purpose and novelty of the present analysis. (This should ideally be in the introduction/background).	project.
--	-----------------

VERSION 2 – REVIEW

REVIEWER	Sarah Cotterill University of Manchester, Centre for Biostatistics
REVIEW RETURNED	04-May-2022

GENERAL COMMENTS	The authors have largely addressed the comments I made, and I am pleased to hear that they found the comments to be helpful. The paper reads well. I just have a few small queries: When reporting trial baseline factors, it is widely recommended that p values are not presented. I suggest that the authors delete the p values column in the table and the corresponding text in the statistical analysis section (p45) and the results section (p48). At the end of the results section the authors have deleted one sentence describing the final analysis. I think they also need to delete the text describing this analysis from the methods section. I understand from their response why the authors choose to retain the cut-points (eg age cut-point at 32), but can they explain in the paper why the particular cut-points were chosen. In the results section the authors state (Page 49) that 'In Model 2, we ... found that each session that a woman attended significantly increased her odds of taking IFA'. This is not quite accurate: all of the categories were compared to zero, rather than to each other, so, while the odds appear to have increased as the number of sessions rose, the differences between the categories cannot be described as 'significant'. The text describing models 3 and 4 may also need adjustment in a similar way. While it is tempting to compare the results by treatment actually received, the danger is that the comparison is then between a compliant group and a non-compliant group, and compliance is often related to the outcome.
---

REVIEWER	Uttara Partap Harvard University T H Chan School of Public Health, Global Health and Population
REVIEW RETURNED	11-Apr-2022

GENERAL COMMENTS	I thank the authors for considering all of the comments provided and responding in so much detail. I think that the manuscript is much improved as a result. I do have some outstanding, mainly minor comments related to further clarifying the revision.
---

	1. Thanks for providing all the detail in Table 1 re: interventions. In Table 1, it seems that there are statements regarding the community-based hemoglobin testing intervention that have been listed under the "Group participatory learning sessions" section: "Testing sessions lasted for an hour as community/group testing was followed up by demonstration of results and behavioral nudges for improving their Hb count." Could the authors please check this is in the appropriate place? 2. Methods section p 12: I think the word "regression" may be missing from the following sentence (inserted in parentheses to demonstrate location): "and obtained p-values testing the null hypothesis of no difference between the two arms via linear and logistic {regression} for continuous and categorical variables, respectively." 3. Addition of detail re: how exposure to participatory learning modules was measured: thank you for doing this in response to my previous comment 2ai. However, I am still slightly unclear on what the variable of group education sessions represents from the text that has been written in the Methods. From the authors' answer to my comment 4, I understand that the number corresponds to the number of times participants report having seen a particular image from any of the group sessions, with this being a proxy for the intensity of exposure to group participatory sessions - is that correct? If so, I think it would be very helpful to state this definition explicitly in the Methods section where the authors define the variable (this to me would be preferable even over the text currently describing the coding for the component variables relating to images and games and how the overall variable was constructed). Otherwise, the impression to readers like myself is that the number represents the number of sessions attended. 4. Note re: the other two variables – please check my understanding and follow-up questions: a. Number of times tested: main outcome variable is technically categorical with values 0, 1, 2, 3, and 4+ (though the authors also note the construction of the continuous variable range 0-4). i. if the continuous variable was used for Table 3 "Average number of hemoglobin tests undergone", I think it might be good to add in a footnote that this continuous variable with a "re-coding" of sorts was used for this measure. This also means that the average number might actually be under-estimated – this might be good to note somewhere – perhaps the results as it is not necessarily a primary finding? b. number of RANI comms videos viewed: variable is a sum of how many of 4 videos seen, so continuous and with a range of 0-4. i. if this is the case, is there any reason why there is a classification of 0, 1, 2, 3, 4+ for the RANI Comms videos in Table 4? I am assuming this is a typographical error and it should be just "4" 5. Thank you for the clarification re: analyses informing Tables 2-4, which I understand are all regression based. Just to note, Table 2 still has a footnote that mentions "T-tests compare demographic differences between treatment versus control arms. " - if the P values presented here are not the results of such a test, then this statement should be removed. 6. If references to communicating with others about the intervention
--	---

	have now been removed, should it still be in the “Independent variables” section, p11?
--	--

VERSION 2 – AUTHOR RESPONSE

Review Comment	Response
Reviewer #1	
When reporting trial baseline factors, it is widely recommended that p values are not presented. I suggest that the authors delete the p values column in the table and the corresponding text in the statistical analysis section (p45) and the results section (p48).	Thank you for this suggestion. We know that there are different opinions about whether to include p values. In our opinion, the p-values add information without taking anything away, so we respectfully decided to include them.
At the end of the results section the authors have deleted one sentence describing the final analysis. I think they also need to delete the text describing this analysis from the methods section.	The analysis description regarding the interaction model is removed.
I understand from their response why the authors choose to retain the cut-points (eg age cut-point at 32), but can they explain in the paper why the particular cut-points were chosen.	We’ve added in a rationale for why we chose the cutoff point.
In the results section the authors state (Page 49) that ‘In Model 2, we ... found that each session that a woman attended significantly increased her odds of taking IFA’. This is not quite accurate: all of the categories were compared to zero, rather than to each other, so, while the odds appear to have increased as the number of sessions rose, the differences between the categories cannot be described as ‘significant’. The text describing models 3 and 4 may also need adjustment in a similar way.	Thank you for pointing this out. We’ve changed the language to your suggestion, and we made similar edits to the description of the results for models 3 and 4.
While it is tempting to compare the results by treatment actually received, the danger is that the comparison is then between a compliant group and a non-compliant group, and compliance is often related to the outcome.	We hope we’ve made this clear in the discussion section. “Furthermore, it is possible that women who participate in the intervention may be more motivated

	to change behavior in general and are already more inclined to take IFA. Therefore, Table 4 could overstate the effects on IFA use of exposure to the intervention components, as well as the contributions of those components to the overall RANI effect. However, IFA use is still a result of participating in the RANI intervention, as participation led to IFA use so we may have simply captured women who were farther along in their readiness to change.”
Reviewer #2	
1. Thanks for providing all the detail in Table 1 re: interventions. In Table 1, it seems that there are statements regarding the community-based hemoglobin testing intervention that have been listed under the "Group participatory learning sessions" section: "Testing sessions lasted for an hour as community/group testing was followed up by demonstration of results and behavioral nudges for improving their Hb count." Could the authors please check this is in the appropriate place?	Thank you. We moved the sentence on testing to the Hb testing description within Table 1.
2. Methods section p 12: I think the word "regression" may be missing from the following sentence (inserted in parentheses to demonstrate location): "and obtained p-values testing the null hypothesis of no difference between the two arms via linear and logistic {regression} for continuous and categorical variables, respectively."	Thank you. Added in.
3. Addition of detail re: how exposure to participatory learning modules was measured: thank you for doing this in response to my previous comment 2ai. However, I am still slightly unclear on what the variable of group education sessions represents from the text that has been written in the Methods. From the authors' answer to my comment 4, I understand that the number corresponds to the number of times participants report having seen a particular image from any of the group sessions, with this being a proxy for the intensity of exposure to group participatory sessions - is that correct? If so, I think it would be very helpful to state this definition explicitly in the	Thank you for this suggestion. We added your suggested text to the measure description, and we feel that its clearer now.

Methods section where the authors define the variable (this to me would be preferable even over the text currently describing the coding for the component variables relating to images and games and how the overall variable was constructed). Otherwise, the impression to readers like myself is that the number represents the number of sessions attended.	
4. Note re: the other two variables – please check my understanding and follow-up questions: a. Number of times tested: main outcome variable is technically categorical with values 0, 1, 2, 3, and 4+ (though the authors also note the construction of the continuous variable range 0-4). i. if the continuous variable was used for Table 3 “Average number of hemoglobin tests undergone”, I think it might be good to add in a footnote that this continuous variable with a “re-coding” of sorts was used for this measure. This also means that the average number might actually be under-estimated – this might be good to note somewhere – perhaps the results as it is not necessarily a primary finding? b. number of RANI comms videos viewed: variable is a sum of how many of 4 videos seen, so continuous and with a range of 0-4. i. if this is the case, is there any reason why there is a classification of 0, 1, 2, 3, 4+ for the RANI Comms videos in Table 4? I am assuming this is a typographical error and it should be just “4”	Regarding the first part of the reviewer’s inquiry, we actually treat the number of times tested variable in three different ways in three different places: an average (in Tables 2 and 3), a dichotomous indicator of being tested at least once (in Table 2), and as four dummy variables representing the five ordered categories (in Table 4). This may seem a bit confusing, but in Tables 2 and 3 our goal was to succinctly summary differences in this variable across study arm and age, education, and caste/tribe group. The mean and proportion both meet the succinctness requirement, albeit in different ways. This is why we used those in Tables 2 and 3. In the regression analysis in Table 4, in contrast, we were not concerned with succinctness but rather wanted to use all of the available information to obtain the maximum possible extent to which the overall treatment effect on IFA use may have been mediated by hemoglobin testing. We therefore used four dummy indicator variables to represent the five ordinal categories for those analyses. Regarding the second part of the reviewer’s inquiry, thank you for pointing that out. We have changed “4+” to simply “4” in Table 4.
5. Thank you for the clarification re: analyses informing Tables 2-4, which I understand are all regression based. Just to note, Table 2 still has a footnote that mentions "T-tests compare demographic differences between treatment versus control arms. " - if the P values presented here are not the results of such a test, then this statement should be removed.	Thank you for bringing this to our attention. We changed the text to: "We ran regression analyses to test the statistical significance between treatment versus control arms on demographic variables. "

6. If references to communicating with others about the intervention have now been removed, should it still be in the “Independent variables” section, p11?	Removed. Thank you.
--	----------------------------